# Relationship between Depressive Symptoms, Caregiver Strain, and Social Support with Dementia Grief in Family Caregivers

**DOI:** 10.3390/medicina60040643

**Published:** 2024-04-17

**Authors:** Miriam Sánchez-Alcón, Almudena Garrido-Fernández, José María Cano-Rojas, José Luis Sánchez-Ramos, Juan Diego Ramos-Pichardo

**Affiliations:** 1Nursing Department, Nursing Faculty, University of Huelva, 21007 Huelva, Spain; almudena.garrido@denf.uhu.es (A.G.-F.); jsanchez@uhu.es (J.L.S.-R.); juan.ramos@denf.uhu.es (J.D.R.-P.); 2Provincial Federation of Associations of Family Caregivers of People with Alzheimer’s Disease and other Dementias of Huelva and Province, 21710 Huelva, Spain; afabollullos@alzheimerdehuelva.org

**Keywords:** dementia grief, family caregiver, depressive symptoms, caregiver strain, social support

## Abstract

*Background and Objectives*: Dementia grief in family caregivers of people with dementia refers to grieving prior to the death of the care recipient. It is related to psychosocial risk factors that may have a negative impact on the health of these family caregivers. This study aimed to describe the relationship between depressive symptoms, caregiver strain, and social support with dementia grief in family caregivers of people with dementia. *Materials and Methods*: A descriptive correlational cross-sectional study was conducted. A total of 250 family caregivers of people with dementia participated. Dementia grief was the main variable, and depressive symptoms, caregiver strain, and social support were assessed. Additionally, socio-demographic data were collected. Descriptive statistics were calculated, and a bivariate correlation analysis and a multiple linear regression analysis were performed for dementia grief. *Results*: Higher scores for dementia grief were found in women, in family caregivers of patients at advanced stages of dementia, and in family caregivers with a low level of education. High levels of depressive symptoms and caregiver strain and low levels of social support indicated greater intensity of dementia grief. Depressive symptomatology was the variable with the greatest influence on dementia grief. Caregiver strain and social support also related to dementia grief, but to a lesser extent. *Conclusions*: In family caregivers, depressive symptoms, caregiver strain, and social support are related to the intensity of dementia grief, with a greater influence of depressive symptoms. Moreover, being female, having a low level of education, and caring for a care recipient at an advanced stage of dementia are factors associated with increased dementia grief. Concerning study limitations, the sample was restricted, belonging to a specific region of Spain and to a Provincial Federation of associations. It is necessary to exercise caution in generalizing results due to the sociodemographic and geographical characteristics of the sample.

## 1. Introduction

Dementia is currently considered a global public health challenge. With the ageing of the world’s population, the prevalence of this disease has been on the rise, with more than 55 million people worldwide affected, and an estimated 139 million affected by 2050 [1].

Dementia not only affects those who suffer from it, but also has a significant impact on the lives of family members who provide care for people with this disease. Caring for people with dementia is a challenging and complex task, given that as the dementia progresses, family caregivers must adapt to the physical and mental deterioration of the care recipient. These changes impose a heavy workload on family caregivers, causing physical, psychological, and social problems that affect their health. In fact, they are often forced to give up a considerable part of their lives to devote to caring, with lifestyle and professional readjustments [2,3].

These conditions are highly stressful and generate a sense of loss in family caregivers prior to the death of the care recipient, which has recently been described as “dementia grief” [4]. Dementia grief is a complex phenomenon that is related to psychosocial and physical variables that can have a negative impact on the quality of life of family caregivers [5,6]. This concept refers to feelings related to the anticipation of future death together with losses (social, professional, emotional, and independence losses) that occur during the experience of caring for people with dementia [4,7,8].

This experience is characterized by progressive and continuous losses caused by the disease; prolonged and uncertain time of care; difficulties in patient-caregiver communication; disappearance of the identity of the loved one, who is physically present but psychologically absent; and deterioration of relationships owing to the new family roles [4,5,9]. Dementia grief is different from anticipatory grief. Anticipatory grief focuses exclusively on the feelings experienced by family caregivers before the death of a loved one occurs [10,11]. However, dementia grief is a broader concept, which also includes the emotional and psychological anticipation of family caregivers prior to the death of the person with dementia, along with the caregiver’s own losses (social, professional, and independence losses, etc.) [6].

A recent review has shown a positive relationship between depression, burden, and social isolation with anticipatory grief in family caregivers of people with dementia. This association suggests that as levels of depression, burden, and social isolation increase, anticipatory grief will also increase. Furthermore, this relationship also indicates that these variables could be considered predictors factors of the onset of anticipatory grief [12]. However, in the literature, there are very few studies linking these variables to dementia grief. 

Some studies have reported a relationship between depressive symptoms [4,13,14,15], strain and overload [8,15,16], and social support [17] in family caregivers with experienced dementia grief. However, few studies have analyzed these variables jointly. Therefore, the aim of this study was to describe the relationship between depressive symptoms, caregiver strain, and social support with dementia grief in family caregivers of people with dementia.

## 2. Materials and Methods

### 2.1. Research Design and Study Participants

A descriptive correlational cross-sectional study was conducted on a sample of 250 family caregivers of people with dementia from the province of Huelva (Spain). For inclusion in this study, the participants had to meet the following inclusion criteria: be at least 18 years of age, be a family caregiver of people diagnosed with dementia in the home and be able to read and speak Spanish. As exclusion criteria, family caregivers with any condition (visual, cognitive, etc.) that may hinder their ability to read and understand were not included.

### 2.2. Data Collection

All family caregivers belonging to the Provincial Federation of Associations of Family Caregivers of People with Alzheimer’s Disease in Huelva who met the inclusion criteria were contacted. Group appointments were scheduled, at which the purpose of the research was made clear to the participants, and a brief description of the research was given to them in printed form. Those who agreed to participate received an informed consent form together with a data collection booklet containing the necessary measurement instruments. Participants completed the questionnaires in approximately 20 min.

### 2.3. Instruments

#### 2.3.1. Marwit–Meuser Caregiver Grief Inventory-Short Form (MM-CGI-SF)

The Marwit–Meuser Caregiver Grief Inventory-Short Form [18] is a tool used to measure dementia grief in caregivers of people with dementia. It consists of 18 items distributed in three subscales, with 6 items each: (a) Personal Sacrifice Burden (PSB), which assesses the personal sacrifices that the caregiver suffers as a consequence of caregiving; (b) Heartfelt Sadness and Longing (HS&L), which measures the emotional responses felt by the caregiver while providing care to the person with dementia; and (c) Worry and Felt Isolation (W&FI), which assesses the caregiver’s perception of the lack of social interaction and support from others.

The items included in the questionnaire were assessed using a 5-point Likert-type response scale, which prompted participants to express their degree of agreement or disagreement with each item (from 1 = Strongly Disagree to 5 = Strongly Agree). The total scores for each subscale were calculated, and additionally, an overall score was obtained by adding the scores of the three subscales. The higher the MM-CGI-SF score, the more intense the grieving experience for the caregiver [19]. 

#### 2.3.2. Patient Health Questionnaire-9 (PHQ-9)

This is a self-administered questionnaire that aims to assess the presence and severity of depressive symptomatology. It has 9 items that participants must answer using a Likert-type scale composed of 4 options, ranging from 0 to 3 points. The overall score of the questionnaire ranges from 0 to 27 points. Its interpretation is as follows: scores between 0 and 4 indicate minimal depressive symptoms; 5 to 9 suggest mild depressive symptoms; 10 to 14 indicate moderate depressive symptoms; 15 to 19 indicate moderately severe depressive symptoms; and a score of 20 to 27 reflects severe depressive symptoms [20,21]. 

#### 2.3.3. Caregiver Strain Index (CSI)

This is a self-assessment questionnaire composed of 13 items with dichotomous responses (true-false). Its purpose is to measure the degree of perceived overload and the level of strain in the performance of the caregiving role of caregivers of severely dependent persons. The total score can vary between 0 and 13 points. A total score equal to or higher than 7 indicates a high level of strain on the part of the caregiver in caring for the dependent person [22,23].

#### 2.3.4. Duke–UNC Functional Social Support Questionnaire

This is a questionnaire that assesses individuals’ perceptions of the assistance and support provided by their family and friends [24,25]. It is a self-administered questionnaire with a structure composed of two dimensions: confidential social support, which addresses the possibility of having people to communicate with; and affective social support, which looks at demonstrations of love, affection, and empathy.

The questionnaire consists of 11 items, rated on a Likert-type response scale ranging from 1 to 5. Therefore, the total scores obtained can vary between 11 and 55 points. A score equal to or higher than 32 indicates perceived fair social support, while a score below 32 suggests perceived low social support.

Socio-demographic data of the participants were collected: age, sex, level of relationship with the care recipient, educational level, days per week caring for the patient, and years of caregiving. In addition, data on the stage of dementia of the care recipient were collected.

### 2.4. Data Analysis

For the description of the sample, descriptive statistics (means, standard deviations, ranges, medians, absolute values, and frequencies) were calculated. Parametric tests were used (Student *t*-test and 1-Way Anova) for the comparison of means.

Pearson correlations were calculated, linking the scores of the MM-CGI-SF and its subscales, and the rest of the variables. Subsequently, a multiple linear regression analysis was performed with the enter method, considering the total MM-CGI-SF and each subscale as dependent variables; in addition, the variables that had shown significant correlations in the correlation analysis were considered independent variables.

A data analysis was carried out using SPSS Statistics v.26 [26], and a 95% confidence level was established to determine statistical significance. The Strengthening the Reporting of Observational studies in Epidemiology (STROBE) guidelines were followed [27]. 

### 2.5. Ethical Aspects

The study was approved by the Huelva Provincial Research Ethics Committee. Throughout the research, anonymity, confidentiality, and an appropriate handling of the participants’ data were guaranteed. The ethical principles and fundamental research standards that govern all scientific research were rigorously maintained, in accordance with the Declaration of Helsinki.

## 3. Results

### 3.1. Descriptive Data of the Sample

A total of 250 family caregivers of people with dementia participated in the study, of whom 80.4% were women, with a mean age of 58.22 (SD = 12.7) years. The majority were daughters of the patients (62%; *n* = 155), with primary education (36%; *n* = 90), who had been caring for their relative presenting with moderate-stage dementia for several years ($\bar{X}$ = 5; SD = 3.6) (69.6%; *n* = 174), with a mean dedication of 6.1 (SD = 1.7) days per week.

With regard to dementia grief, the mean total score was 64.6 (SD = 14.8). Among the dimensions of the dementia grief, the W&FI scored the lowest ($\bar{X}$ = 18.5, SD = 5.7), and the HS&L, the highest ($\bar{X}$ = 23.3, SD = 5.4). The mean score for the PHQ-9 questionnaire was 11 (SD = 7); for the CSI, it was 7 (SD = 3.1); and for the Duke-UNC, it was 37.3 (SD = 10.4). Table 1 shows the descriptive data of the sample.

Table 2 shows the mean scores of the dementia grief and its dimensions in relation to the rest of the variables. Differences were found in regards to sex, with higher scores in women, as well as regarding the educational level of the family caregiver, with higher scores in family caregivers with no or incomplete education, and in relation to the stage of dementia of the care recipient, with higher scores in cases of more advanced stages of dementia.

Differences were also observed in dementia grief scores as regards depressive symptoms and caregiver strain, with higher scores in participants with more depressive symptoms and high caregiver strain. In terms of social support, the intensity of grief was higher in family caregivers with low social support.

### 3.2. Bivariate Correlations

Table 3 shows the correlations linking the MM-CGI-SF and its subscales, and the rest of the variables. Statistically significant positive correlations were found with depressive symptoms and caregiver strain, indicating a direct relationship. The correlation with perceived social support was also significant, although, in this case, with a negative value indicating an indirect relationship (the better the perceived social support, the lower the score in the MM-CGI-SF).

With respect to the socio-demographic variables, the sex and educational level of the family caregiver, the stage of dementia in which the care recipient was, the family relationship between both of them, and the time devoted to caregiving were the variables that showed a relationship with the MM-CGI-SF (Table 3).

### 3.3. Multiple Linear Regression Analysis

Table 4 shows the multiple linear regression analysis results, considering total dementia grief and its three dimensions as dependent variables. For total dementia grief, the variable with the greatest weight was depressive symptomatology, followed by caregiver strain, stage of dementia, and educational level. 

As regards PSB, caregiver strain was the variable with the greatest weight, followed by depressive symptoms, weekly days of care, and stage of dementia. Regarding the HS&L model, depressive symptoms had the greatest weight, followed by stage of dementia, caregiver strain, and educational level.

In the linear regression analysis for the W&FI, the greatest weight was also found for depressive symptoms, followed by caregiver strain, social support, educational level, and weekly days of care. The explained variance for the W&FI model was higher (50.2%) than for the PSB, HS&L, and total dementia grief regression models (44.7%, 31.2%, and 49.6%, respectively).

## 4. Discussion

This study aimed to describe the relationship between depressive symptoms, caregiver strain, and social support with dementia grief in family caregivers of people with dementia.

In terms of socio-demographic variables, being female, having a low educational level, and caring for a care recipient who is at an advanced stage of the disease seem to be related to an increase in dementia grief.

Most of the family caregivers were women, and they had higher dementia grief scores than men. This may be caused because women are also often involved in other tasks, such as child and household care [28,29]. In addition, these caregivers lack free time and often limit their employment to part-time work or even sacrifice their job and opportunities for advancement, affecting family income and financial stability. In this sense, the family’s finances fall on the spouse or partner, increasing the economic vulnerability of the household and the economic dependence of women on their partners [30,31,32]. Although there are studies showing that women experience more dementia grief [33,34], other articles have shown no association between sex and the level of dementia grief [16,35], showing that dementia grief may be more associated with individual factors (personality, psychosocial factors, etc.) rather than sex. 

Family caregivers with higher educational levels experienced a lower intensity of dementia grief. This may be because they had more personal resources such as information, social contacts, and familiarity with the regulations, and very probably also greater economic resources. These results align with the findings of several studies [36,37,38]. With regard to the stage of dementia of the care recipient, higher dementia grief scores were observed for the family caregiver, especially when the care recipient was at an advanced stage of the disease. This association could be linked to an increase in the demands for care of the patient as the dementia progresses [12,39], as well as to the accumulation of care time when reaching these advanced stages, which can already be years or even decades [33,37,40].

Regarding the main variables of this study, it was observed that high levels of depressive symptoms and caregiver strain and low levels of social support indicated greater intensity of dementia grief.

Based on our findings, depressive symptomatology was the variable that showed the most weight and influence on dementia grief. Depressive symptoms are associated with deep feelings of sadness, hopelessness, and loss. They occur frequently in family caregivers of people with dementia, especially when they are aware that there will be no improvement for the care recipient. These feelings affect the family caregiver’s motivation and enthusiasm for life [41,42]. Moreover, if even before the death of the recipient, family caregivers show depressive symptoms, once the death occurs, the risk of complicated grief or depression is much higher [43]. Our results are consistent with numerous studies linking depressive symptoms with increased dementia grief [4,13,14,15].

As for caregiver strain, it was observed that it could be a predictor of dementia grief. As expected, for the Personal Sacrifice Burden (PSB) dimension, caregiver strain was the variable with the greatest influence, as it precisely assesses this dimension. Caring for a person with dementia can be physically and emotionally demanding, causing family caregivers fatigue, stress, and emotional exhaustion. They often live tired due to the continuity of care, go through different stages of dementia for years, and are constantly confronted with the increasing needs of care recipients [44,45,46]. All of these factors may increase the sense of strain, and may therefore intensify dementia grief. These data are consistent with previous research showing that strain is associated with higher levels of pre-death grief in family caregivers of people with dementia [8,16,34,47].

The study findings also show that family caregivers with low perceived social support experienced a higher intensity of dementia grief. Furthermore, the variable with the highest weight in the Worry and Felt Isolation (W&FI) dimension was depressive symptomatology and not social support as we might expect. This may be because depressive symptoms cause the family caregiver to withdraw socially, gradually reducing their social circle, feeling less emotionally connected to others, and having a decreasing sense of social support. If this situation persists over the years of caregiving, it may even lead to the isolation of the family caregiver [12,48]. Recent studies [8,17,33] are in line with our data, indicating that social support may act as a buffer on the severity of dementia grief.

Multiple regression analyses showed that depressive symptoms, caregiver strain, and social support influence the intensity of dementia grief, with depressive symptomatology being the variable with the greatest influence on dementia grief and on two of its dimensions, Heartfelt Sadness and Longing (HS&L) and Worry and Felt Isolation (W&FI).

Therefore, there is a need for social and health professionals to assess the mood, strain, and perceived social support of family caregivers of people with dementia to identify individuals at risk and develop interventions aimed at preventing complicated grief after the death of the care recipient.

There are limitations to this study that should be mentioned. On the one hand, the sample size was insufficient for drawing a conclusion. However, it is important to take into consideration that the study is based on caregivers of people with dementia who face a significant burden of responsibilities. These caregivers are fully dedicated to the care of their loved ones and have very little time to devote to other activities, such as participation in research studies.

On the other hand, the sample was limited in geographical terms, since the data collection focused exclusively on the province of Huelva (Spain). Therefore, it is necessary to exercise caution when interpreting and generalizing the results, considering the sociodemographic and geographical characteristics of the sample. However, there are no significant cultural differences among the different geographical regions of Spain, and family caregivers joined associations for the services they offered, regardless of the caregivers’ economic or educational resources. This leads us to consider that our results may offer a close representation of the situation of Spanish family caregivers.

It is noteworthy that the study sample consisted exclusively of family caregivers who belonged to the Huelva Provincial Federation of Associations of Family Caregivers of People with Alzheimer’s Disease, so it was not possible to recruit family caregivers who were not members of this association. The analysis of this particular group of family caregivers may have had a mitigating effect with regard to dementia grief scores, as the family caregivers who belong to this association have greater access to services. However, the results show that these family caregivers scored high in caregiver strain and low in perceived social support, thus influencing the intensity of dementia grief. Therefore, the inclusion of non-associated family caregivers in the sample would not have substantially altered the results and conclusions of this study.

## 5. Conclusions

In family caregivers of people with dementia, experiencing depressive symptoms and having a high strain and low social support are related to the intensity of dementia grief. According to our data, the variable with the greatest weight on dementia grief is depressive symptomatology. Moreover, being female, having a low level of education, and caring for a care recipient at an advanced stage of dementia are factors associated with increased dementia grief. Therefore, it is necessary to carry out assessments and interventions that consider these variables to prevent complicated grief after the death of the care recipient. 

## Figures and Tables

**Table 1 medicina-60-00643-t001:** Descriptive data of the sample.

Variables	*n* (%)	Mean	SD	Range	Median
Age		58.2	12.7	24–87	57
Sex					
Female	201 (80.4%)				
Male	49 (19.6%)				
Relationship with the care recipient					
Spouse/partner	79 (31.6%)				
Son/daughter	155 (62%)				
Other	16 (6.4%)				
Educational level					
No or incomplete education	29 (11.6%)				
Primary education	90 (36%)				
Secondary education	68 (27.2%)				
University education	53 (21.2%)				
Postgraduate education	10 (4%)				
Stage of dementia					
Mild	51 (20.4%)				
Moderate	174 (69.6%)				
Severe	25 (10%)				
Weekly days of care		6.1	1.7	1–7	7
Years of care		5	3.6	1–20	4
Personal Sacrifice Burden MM-CGI-SF(PSB)		22.7	5.4	6–30	23
Heartfelt Sadness and Longing MM-CGI-SF(HS&L)		23.3	5.4	6–30	25
Worry and Felt Isolation MM-CGI-SF(W&FI)		18.5	5.7	6–30	18
Dementia grief Total MM-CGI-SF		64.6	14.8	18–90	67
Depressive symptoms (PHQ-9)		11	7	0–27	10
Minimal	54 (21.6%)				
Mild	63 (25.2%)				
Moderate	58 (23.2%)				
Moderately severe	37 (14.8%)				
Severe	38 (15.2%)				
Caregiver Strain (CSI)		7	3.1	0–13	7
No high	105 (42%)				
High	145 (58%)				
Functional Social Support (Duke-UNC)		37.3	10.4	11–55	38
Low	69 (27.6%)				
Normal	181 (72.4%)				

SD: Standard deviation; MM-CGI-SF: Marwit-Meuser Caregiver Grief Inventory-Short Form; MM-CGI-SF(PSB): Marwit-Meuser Caregiver Grief Inventory-Short Form (Subescale Personal Sacrifice Burden); MM-CGI-SF(HS&L): Marwit-Meuser Caregiver Grief Inventory-Short Form (Subescale Heartfelt Sadness and Longing); MM-CGI-SF(W&FI): Marwit-Meuser Caregiver Grief Inventory-Short Form (Subescale Worry and Felt Isolation); PHQ-9: Patient Health Questionnaire-9; CSI: Caregiver Strain Index; Duke-Unc: Duke–UNC Functional Social Support Questionnaire.

**Table 2 medicina-60-00643-t002:** Mean scores of the total MM-CGI-SF and its subscales in relation to the rest of the variables.

Variables	Personal Sacrifice Burden MM-CGI-SF(PSB)		Heartfelt Sadness and Longing MM-CGI-SF(HS&L)		Worry and Felt Isolation MM-CGI-SF(W&FI)		Dementia Grief Total MM-CGI-SF	
	Mean (SD)	*t*/*F* (*p*)	Mean (SD)	*t*/*F* (*p*)	Mean (SD)	*t*/*F* (*p*)	Mean (SD)	*t*/*F* (*p*)
Sex								
Female	23.1 (5.1)	–1.745 (0.056)	23.7 (5)	−2.057 (0.044)	18.9 (5.6)	–2.185 (0.030)	65.7 (14.1)	–2.439 (0.015)
Male	21.3 (6.3)		21.6 (6.4)		16.9 (5.8)		60 (16.7)	
Relationship with the care recipient								
Spouse/partner	23.6 (5.5)	2.075 (0.104)	24.2 (5.6)	1.401 (0.243)	19.6 (5.9)	1.972 (0.119)	67.6 (15.3)	2.140 (0.096)
Son/daughter	22.4 (5.4)		22.9 (5.4)		18.1 (5.6)		63.5 (14.8)	
Other	19.8 (4.5)		21.6 (4.7)		16.2 (4.2)		57.6 (10.8)	
Educational level								
No or incomplete education	25.3 (5.5)	3.068 (0.017)	26.4 (3.8)	4.983 (0.001)	22.7 (5.4)	6.459 (0.000)	74.5 (13.6)	5.793 (0.000)
Primary education	22.8 (5.1)		23.1 (5.7)		18.5 (5.4)		64.5 (14.4)	
Secondary education	22.5 (5.1)		23.5 (5.2)		18.4 (5.7)		64.4 (14.5)	
University education	22.2 (5.9)		22.5 (4.9)		16.9 (5.1)		61.7 (14.2)	
Postgraduate education	19.1 (5)		18.5 (5.9)		14.9 (5.3)		52.5 (13.7)	
Stage of dementia								
Mild	19.6 (6.5)	13.179 (0.000)	20.2 (6.4)	13.489 (0.000)	16.9 (6.6)	2.549 (0.080)	56.8 (18.2)	10.641 (0.000)
Moderate	23.2 (4.9)		23.8 (4.9)		18.8 (5.4)		66.0 (13.5)	
Severe	25.4 (3.7)		26 (3.7)		19.2 (5.3)		70.6 (9.7)	
Depressive symptoms (PHQ-9)								
Minimal	17.3 (5.3)	29.300 (0.000)	18.9 (6.2)	17.467 (0.000)	13.1 (4.2)	41.644 (0.000)	49.4 (14.1)	37.967 (0.000)
Mild	22.8 (4.2)		22.9 (4.9)		17.4 (4.5)		63.2 (11.4)	
Moderate	23.7 (4.8)		24.2 (4.4)		18.7 (4.7)		66.7 (12.4)	
Moderately severe	25.3 (3.8)		25.8 (3.8)		23.1 (4.1)		74.3 (9.5)	
Severe	26.4 (3.6)		26.1 (3.6)		23.2 (4.1)		75.8 (9.3)	
Caregiver Strain (CSI)								
No high	19.9 (5.9)	−7.283 (0.000)	21.5 (6.1)	–4.315 (0.000)	16.1 (5.8)	–6.132 (0.000)	57.6 (16)	–6.570 (0.000)
High	24.8 (3.9)		24.6 (4.4)		20.2 (4.9)		69.7 (11.5)	
Functional Social Support (Duke-UNC)								
Low	24.6 (4.6)	3.404 (0.001)	24.6 (4.8)	2.344 (0.020)	21.7 (5.2)	5.886 (0.000)	71 (13.4)	4.336 (0.000)
Normal	22.0 (5.5)		22.8 (5.5)		17.2 (5.4)		62.1 (14.6)	

*t*: Student t-test for independent samples; F: 1-Way Anova; *p*: significance level; SD: Standard deviation; MM-CGI-SF: Marwit-Meuser Caregiver Grief Inventory-Short Form; MM-CGI-SF(PSB): Marwit-Meuser Caregiver Grief Inventory-Short Form (Subescale Personal Sacrifice Burden); MM-CGI-SF(HS&L): Marwit-Meuser Caregiver Grief Inventory-Short Form (Subescale Heartfelt Sadness and Longing); MM-CGI-SF(W&FI): Marwit-Meuser Caregiver Grief Inventory-Short Form (Subescale Worry and Felt Isolation); PHQ-9: Patient Health Questionnaire-9; CSI: Caregiver Strain Index; Duke-Unc: Duke–UNC Functional Social Support Questionnaire.

**Table 3 medicina-60-00643-t003:** Bivariate correlations between the MM-CGI-SF and its subscales in relation to the rest of the variables.

Variables	Personal Sacrifice Burden MM-CGI-SF(PSB)	Heartfelt Sadness and Longing MM-CGI-SF(HS&L)	Worry and Felt Isolation MM-CGI-SF(W&FI)	Dementia Grief Total MM-CGI-SF
Age	0.103 *p* = 0.104	0.075 *p* = 0.238	0.110 *p* = 0.082	0.107 *p* = 0.90
Sex	0.125 *p* = 0.048	0.149 *p* = 0.018	0.137 *p* = 0.030	0.153 *p* = 0.015
Relationship with the care recipient	−0.132 *p* = 0.038	−0.115 *p* = 0.069	−0.149 *p* = 0.018	−0.148 *p* = 0.020
Educational level	−0.181 *p* = 0.004	−0.210 *p* = 0.001	−0.270 *p* < 0.001	−0.247 *p* < 0.001
Stage of dementia	0.304 *p* < 0.001	0.308 *p* < 0.001	0.129 *p* = 0.041	0.273 *p* < 0.001
Weekly days of care	0.243 *p* < 0.001	0.116 *p* = 0.067	0.220 *p* < 0.001	0.216 *p* = 0.001
Years of care	0.180 *p* = 0.004	0.174 *p* = 0.006	0.114 *p* = 0.071	0.173 *p* = 0.006
Personal Sacrifice Burden MM-CGI-SF(PSB)		0.727 *p* < 0.001	0.720 *p* < 0.001	0.908 *p* < 0.001
Heartfelt Sadness and Longing MM-CGI-SF(HS&L)	0.727 *p* < 0.001		0.673 *p* < 0.001	0.890 *p* < 0.001
Worry and Felt Isolation MM-CGI-SF(W&FI)	0.720 *p* < 0.001	0.673 *p* < 0.001		0.893 *p* < 0.001
Dementia grief Total MM-CGI-SF	0.908 *p* < 0.001	0.890 *p* < 0.001	0.893 *p* < 0.001	
Depressive symptoms (PHQ-9)	0.537 *p* < 0.001	0.457 *p* < 0.001	0.640 *p* < 0.001	0.609 *p* < 0.001
Caregiver Strain (CSI)	0.497 *p* < 0.001	0.379 *p* < 0.001	0.444 *p* < 0.001	0.490 *p* < 0.001
Functional Social Support (Duke-UNC)	−0.354 *p* < 0.001	−0.289 *p* < 0.001	−0.477 *p* < 0.001	−0.418 *p* < 0.001

*p*: significance level; MM-CGI-SF: Marwit-Meuser Caregiver Grief Inventory-Short Form; MM-CGI-SF(PSB): Marwit-Meuser Caregiver Grief Inventory-Short Form (Subescale Personal Sacrifice Burden); MM-CGI-SF(HS&L): Marwit-Meuser Caregiver Grief Inventory-Short Form (Subescale Heartfelt Sadness and Longing); MM-CGI-SF(W&FI): Marwit-Meuser Caregiver Grief Inventory-Short Form (Subescale Worry and Felt Isolation); PHQ-9: Patient Health Questionnaire-9; CSI: Caregiver Strain Index; Duke-Unc: Duke–UNC Functional Social Support Questionnaire.

**Table 4 medicina-60-00643-t004:** Multiple linear regression results for each dependent variable.

Dependent Variables	Independent Variables	*B*	*SE*	*β*	*t*	*p*	Adjusted R^2^
Personal Sacrifice Burden MM-CGI-SF (PSB)							0.447
	**Constant**	10.538	2.593		4.064	*<0.001*	
	**Caregiver Strain (CSI)**	0.588	0.98	0.344	6.010	*<0.001*	
	**Depressive symptoms (PHQ-9)**	0.235	0.046	0.302	5.065	*<0.001*	
	**Weekly days of care**	0.546	0.166	0.171	3.281	0.001	
	**Stage of dementia**	1.518	0.522	0.151	2.909	0.004	
Heartfelt Sadness and Longing MM-CGI-SF(HS&L)							0.312
	**Constant**	15.275	2.456		6.219	*<0.001*	
	**Depressive symptoms (PHQ-9)**	0.242	0.050	0.312	4.876	*<0.001*	
	**Stage of dementia**	2.045	0.572	0.204	3.577	*<0.001*	
	**Caregiver Strain (CSI)**	0.307	0.105	0.180	2.936	0.004	
	**Educational level**	−0.725	0.280	−0.141	−2.590	0.010	
Worry and Felt Isolation MM-CGI-SF(W&FI)							0.502
	**Constant**	15.037	2.548		5.902	*<0.001*	
	**Depressive symptoms (PHQ-9)**	0.325	0.046	0.398	7.078	*<0.001*	
	**Caregiver Strain (CSI)**	0.408	0.097	0.228	4.211	*<0.001*	
	**Functional Social Support (Duke-UNC)**	−0.095	0.029	−0.174	−3.319	0.001	
	**Educational level**	−0.785	0.258	−0.145	−3.046	0.003	
	**Weekly days of care**	0.339	0.165	0.101	2.049	0.042	
Dementia grief Total MM-CGI-SF							0.496
	**Constant**	48.117	5.821		8.266	*<0.001*	
	**Depressive symptoms (PHQ-9)**	0.824	0.120	0.388	6.877	*<0.001*	
	**Caregiver Strain (CSI)**	1.206	0.249	0.258	4.849	*<0.001*	
	**Stage of dementia**	3.965	1.342	0.145	2.955	0.003	
	**Educational level**	−2.008	0.665	−0.142	−2.951	0.003	

*B*: Non-standardized regression coefficient; *SE*: standard error of the estimate; *β*: standardized regression coefficients; *t*: Student’s test; *p*: significance level; MM-CGI-SF: Marwit-Meuser Caregiver Grief Inventory-Short Form; MM-CGI-SF(PSB): Marwit-Meuser Caregiver Grief Inventory-Short Form (Subescale Personal Sacrifice Burden); MM-CGI-SF(HS&L): Marwit-Meuser Caregiver Grief Inventory-Short Form (Subescale Heartfelt Sadness and Longing); MM-CGI-SF(W&FI): Marwit-Meuser Caregiver Grief Inventory-Short Form (Subescale Worry and Felt Isolation); PHQ-9: Patient Health Questionnaire-9; CSI: Caregiver Strain Index; Duke-Unc: Duke–UNC Functional Social Support Questionnaire.

## Data Availability

Data are available upon request from the Nursing Department, University of Huelva, by contacting the first author, Miriam Sánchez-Alcón: miriam.sanchez@denf.uhu.es.

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
