# Peer review of "Relationship between Depressive Symptoms, Caregiver Strain, and Social Support with Dementia Grief in Family Caregivers"

_medicina, 2024, doi:10.3390/medicina60040643_

Round 1

Reviewer 1 Report

Comments and Suggestions for Authors

The study tries to find a relationship between depressive symptoms, caregiver strain, and social support with dementia grief in family caregivers of people with dementia. The study is not innovative; it just adds to the known literature about the relationship between depressive symptoms, carer strain, and social support with dementia sorrow in family carers of people with dementia in Spanish communities.

The small sample size is insufficient to draw a conclusion. The large difference between included male (49) and female (201) can lead to bias and inaccurate results.

How can the authors validate and normalize patients’ response?Lack of another psychometrically sound instrument assessing unique grief reactions of AD family caregivers might have created a disadvantage while establishing concurrent validity of the scale.

The conclusion is consistent with the evidence and the reported results. However, the small sample size is insufficient to draw a conclusion. 3-           Demographic features of the sample, such as geographic location and educational environment, may impact the generalizability of the findings.

The reference is appropriate.

This study is interesting, however there are some factors to consider.

1-      The small sample size is insufficient to draw a conclusion.

2-      The large difference between included male (49) and female (201) can lead to bias and inaccurate results.

3-      Demographic features of the sample, such as geographic location and educational environment, may impact the generalizability of the findings.

4-      How can the authors validate and normalize patients’ response?

5-      Lack of another psychometrically sound instrument assessing unique grief reactions of AD family caregivers might have created a disadvantage while establishing concurrent validity of the scale.

6-      The authors should include study limitations in the abstract.

Author Response

RESPONSE TO REVIEWER

Dear Reviewer,

Thank you for your interest in our manuscript medicina-2937528 "Relationship between depressive symptoms, caregiver strain, and social support with dementia grief in family caregivers”. We sincerely thank for the editorial and peer review comments that have allowed us to improve the quality of the draft.

We have attached the modified draft and we respond here to the suggestions, including specific references to alterations in the text.

Yours sincerely.

COMMENT FROM REVIEWER: The study tries to find a relationship between depressive symptoms, caregiver strain, and social support with dementia grief in family caregivers of people with dementia. The study is not innovative; it just adds to the known literature about the relationship between depressive symptoms, carer strain, and social support with dementia sorrow in family carers of people with dementia in Spanish communities.

Thank you very much for your comment. Dementia grief is a novel concept, much broader than anticipatory grief. Dementia grief includes the emotional and psychological anticipation of family caregivers prior to the death of the person with dementia, along with the caregiver's own losses (social, professional, loss of independence, etc.).

As this concept is recent, there are few studies in the literature that explore the relationship between psychosocial variables and dementia grief. The few published studies have examined these variables (depressive symptoms, strain caregiver, and social support) individually with dementia grief. However, there is a lack of research that analyses these variables as a whole, relating them to dementia grief. Therefore, we consider our study innovative, as it focuses on describing how depressive symptoms, caregiver strain and social support relate with dementia grief.

COMMENT FROM REVIEWER: The small sample size is insufficient to draw a conclusion. The large difference between included male (49) and female (201) can lead to bias and inaccurate results.

Thank you for your comment. With regard to the sample size, we admit that the sample size is insufficient to draw a conclusion.

However, it is important to take into consideration that the study is based on caregivers of people with dementia who face a significant burden of responsibilities. These caregivers are fully dedicated to the care of their loved ones and have very little time to devote to other activities, such as participation in research studies. This sample was all we were able to achieve, as the participants belong to associations in the province of Huelva. Despite this, our sample finally consisted of 250 family caregivers who agreed to participate in the research and who met the inclusion criteria of the study.

Even so, we have added in the limitations section the following sentences (page 11, lines 304-309):

"On the one hand, the sample size is insufficient to draw a conclusion. However, it is important to take into consideration that the study is based on caregivers of people with dementia who face a significant burden of responsibilities. These caregivers are fully dedicated to the care of their loved ones and have very little time to devote to other activities, such as participation in research studies."

On the other hand, our results show that the majority of caregivers of people with dementia were women. These data are in line with a large number of studies. Here we can refer to several recent studies:

Rosende-Roca, M., Cañabate, P., Moreno, M., Preckler, S., Seguer, S., Esteban, E., Tartari, J. P., Vargas, L., Narvaiza, L., Pytel, V., Bojaryn, U., Alarcon, E., González-Pérez, A., Gurruchaga, M. J., Tárraga, L., Ruiz, A., Marquié, M., Boada, M., & Valero, S. (2022). Sex, Neuropsychiatric Profiles, and Caregiver Burden in Alzheimer’s Disease Dementia: A Latent Class Analysis. Journal of Alzheimer’s Disease, 89(3), 993. https://doi.org/10.3233/JAD-215648

Tchounwou, P. B., Tomata, Y., Huijsman, R., Frías, C. E., Casafont, C., Cabrera, E., & Zabalegui, A. (2022). Validation of the Spanish Version of the Double Knowledge Expectations and Received Knowledge Significant Other Scale for Informal Caregivers of People with Dementia (KESO-DEM/RKSO-DEM). International Journal of Environmental Research and Public Health, 19(9), 5314. https://doi.org/10.3390/IJERPH19095314

Ruisoto, P., Contador, I., Fernández-Calvo, B., Serra, L., Jenaro, C., Flores, N., Ramos, F., & Rivera-Navarro, J. (2020). Mediating effect of social support on the relationship between resilience and burden in caregivers of people with dementia. Archives of Gerontology and Geriatrics, 86, 103952. https://doi.org/10.1016/J.ARCHGER.2019.103952

Gumikiriza-Onoria, J. L., Nakigudde, J., Giordani, B., Mayega, R. W., Sajatovic, M., Mukasa, M. K., Buwembo, D., Lwere, K., & Nakasujja, N. (2024). Psychological Distress among family caregivers of persons with Alzheimer’s Disease and Related Dementias in Uganda. Research Square. https://doi.org/10.21203/RS.3.RS-3918857/V1

Kim, E. A., Shin, S. S., & Lee, J. A. (2024). Relationship Between Acculturation and Mental Health in Korean American Family Caregivers of Community-Dwelling Persons Living with Dementia. Clinical Nursing Reserach. https://doi.org/10.1177/10547738241235695

In Spain, traditionally, due to social and cultural factors, women have assumed caring roles in the home. In caring for people with dementia, it is usually a woman who is the main caregiver, who is responsible for the care of the person with dementia who is ill. As the literature shows, the majority of caregivers are women. The results presented are aligned with reality and capture the most representative experiences of those directly involved in caring for people with dementia.

In addition, by focusing on the experiences of caregivers, these studies can provide valuable information about the specific challenges these women face in their caregiving, which in turn can guide the development of more useful and effective policies and support programmes. While it is important to be aware of the possibility of bias in research, the predominance of female caregivers does not necessarily lead to inaccurate results; rather, it provides a more realistic and accurate perspective on dementia care.

COMMENT FROM REVIEWER: How can the authors validate and normalize patients’ response? Lack of another psychometrically sound instrument assessing unique grief reactions of AD family caregivers might have created a disadvantage while establishing concurrent validity of the scale.

Thank you very much for your contribution. We consider that the objective of this study is to describe the relationships between variables and not to evaluate the metric properties of the questionnaire. Therefore, each variable has been evaluated with a single instrument and not with several.

Dementia grief is a relatively new and specific concept of study, and for its assessment, there are a very limited number of tools available, as Dehpour and Koffman describe in their study:

Dehpour, T., & Koffman, J. (2022). Assessment of anticipatory grief in informal caregivers of dependants with dementia: a systematic review. Aging & Mental Health, 27(1), 110–123. https://doi.org/10.1080/13607863.2022.2032599

In the Spanish population, only the Marwit-Meuser Caregiver Grief Inventory-Short Form (MM-CGI-SF) is available. This Spanish version had adequate psychometric properties. It showed good reliability, with Cronbach's alpha coefficient for the total instrument of 0.927; and for each subscale between 0.822 and 0.854. Additionally, convergent validity was confirmed with related measures (The Patient Health Questionnaire-9, Caregiver Strain Index, Duke–UNC Functional Social Support Questionnaire, and the WHOQOL-BREF questionnaire), and construct validity was assessed through confirmatory factor analysis. This Spanish version is published in the following study:

Sánchez‐Alcón, M., Sosa‐Cordobés, E., Garrido‐Fernández, A., Sánchez‐Ramos, J. L., & Ramos‐Pichardo, J. D. (2023). Psychometric properties of a Spanish version of the MM‐CGI‐SF in caregivers of people with dementia. Journal of the American Geriatrics Society, 1–8. https://doi.org/10.1111/jgs.18623

It is important to consider the specific context of the research. Since there is no other instrument that assesses dementia grief on the Spanish population, concurrent validation could not be studied. However, other evidences of validity, such as construct validity and convergent validity, was used, providing a solid basic for the validity of the scale.

COMMENT FROM REVIEWER: The conclusion is consistent with the evidence and the reported results. However, the small sample size is insufficient to draw a conclusion. 3- Demographic features of the sample, such as geographic location and educational environment, may impact the generalizability of the findings.

Thank you very much for your contribution. We agree with your assessment. For this reason, we have added in limitations of the study the following sentences (page 11, lines 304-309):

"On the one hand, the sample size is insufficient to draw a conclusion. However, it is important to take into consideration that the study is based on caregivers of people with dementia who face a significant burden of responsibilities. These caregivers are fully dedicated to the care of their loved ones and have very little time to devote to other activities, such as participation in research studies."

With regard to the demographic characteristics of the sample, we consider your comment and these are limitations of the study. Therefore, we have added the following sentences to the discussion section (page 11, lines 310-317):

“On the other hand, the sample was limited in geographical terms, since the data collection focused exclusively on the province of Huelva (Spain). Therefore, it is necessary to exercise caution when interpreting and generalizing the results, considering the sociodemographic and geographical characteristics of the sample. However, there are no significant cultural differences among the different geographical regions of Spain, and family caregivers joined associations for the services they offered, regardless of the caregivers' economic or educational resources. This leads us to consider that our results may offer a close representation of the situation of Spanish family caregivers”.

COMMENT FROM REVIEWER: The reference is appropriate.

Thank you very much for your comment.

COMMENT FROM REVIEWER: The authors should include study limitations in the abstract.

Thank you for your comment. We have added the following sentences to the summary, taking into account the maximum of 300 words (page 1, lines 27-30):

“As study limitations, the sample is restricted, belonging to a specific region of Spain and to a Provincial Federation of associations. It is necessary to exercise caution in generalizing results due to the sociodemographic and geographical characteristics of the sample”.

Reviewer 2 Report

Comments and Suggestions for Authors

Dear Authors, kindly use the caregivers instead of family members. In developed countries, family support is lacking. Please work on the introduction section and use appropriate references for defining "Dementia Grief".

Please explain the kind of relationship either positive or negative between depression, burden, and social isolation with anticipatory grief in family caregivers of people with dementia. 

Kindly include the approval number in the methods section for line- The study was approved by the Huelva Provincial Research Ethics Committee. 80% of females are caregivers in your study and how it biased the results in terms of the financial stability of the family. please elaborate on the discussion. Authors are also requested to provide data on the socio-economic status of families in this sample population. 

Comments on the Quality of English Language

Needs mild improvement in the flow of the manuscript.

Author Response

RESPONSE TO REVIEWER

Dear Reviewer,

Thank you for your interest in our manuscript medicina-2937528 "Relationship between depressive symptoms, caregiver strain, and social support with dementia grief in family caregivers”. We sincerely thank for the editorial and peer review comments that have allowed us to improve the quality of the draft.

We have attached the modified draft and we respond here to the suggestions, including specific references to alterations in the text.

Yours sincerely.

COMMENT FROM REVIEWER: Dear Authors, kindly use the caregivers instead of family members. In developed countries, family support is lacking. Please work on the introduction section and use appropriate references for defining "Dementia Grief".

Thank you very much for your comment. In Spain, unlike in other developed countries, there are still many families who take care of people with dementia. Moreover, one of the inclusion criteria for participants in this study was to be a family caregiver of people diagnosed with dementia. For this reason, we continually use the term “family caregivers”.

About the introduction, we have modified the description of “Dementia grief” and the bibliographic references in the manuscript (page 2, lines 45-51):

“These conditions are highly stressful and generate in family caregivers a sense of loss prior to the death of the care recipient, which has recently been described as "dementia grief" [4]. Dementia grief is a complex phenomenon that is related to psychosocial and physical variables that can have a negative impact on the quality of life of family caregivers [5,6]. This concept refers to feelings related to anticipation of future death together with losses (social, professional, emotional, independence losses) that occur during the experience of caring for people with dementia [4,7,8].”

We have modified the bibliographical references in the introduction (page 2, line 56): “[4,5,9]”.  

The bibliography has been revised and reorganized in a different way:

“4. Blandin, K.; Pepin, R. Dementia Grief: A Theoretical Model of a Unique Grief Experience. Dementia 2017, 16, 67–78. https://doi.org/10.1177/1471301215581081.

  1. Crawley, S.; Sampson, E.L.; Moore, K.J.; Kupeli, N.; West, E. Grief in Family Carers of People Living with Dementia: A Systematic Review. Int Psychogeriatr 2022, 1–32, https://doi.org/10.1017/S1041610221002787
  2. Arruda, E.H.; Paun, O. Dementia Caregiver Grief and Bereavement: An Integrative Review. West J Nurs Res 2017, 39, 825–851. https://doi.org/10.1177/0193945916658881

COMMENT FROM REVIEWER: Please explain the kind of relationship either positive or negative between depression, burden, and social isolation with anticipatory grief in family caregivers of people with dementia.

Thank you very much for your contribution. We have modified the introduction section by presenting the relationship between depression, burden and social isolation with anticipatory grief in family caregivers of people with dementia (page 2, lines 62-66):

“A recent review has shown a positive relationship between depression, burden and social isolation with anticipatory grief in family caregivers of people with dementia. This association suggests that as levels of depression, burden and social isolation increase, anticipatory grief will also increase. Furthermore, this relationship also indicates that these variables could be considered predictors factors of the onset of anticipatory grief [12]”. 

COMMENT FROM REVIEWER: Kindly include the approval number in the methods section for line. The study was approved by the Huelva Provincial Research Ethics Committee.

Thank you very much for your comment. Indeed, the study was approved by the Huelva Provincial Research Ethics Committee. The authors have the official document that was given to us, which shows its approval. However, this approval does not have a specific number. In case you want to verify the approval from the ethics committee, the authors are willing to attach the resolution to the journal platform as a supplementary file.

COMMENT FROM REVIEWER: 80% of females are caregivers in your study and how it biased the results in terms of the financial stability of the family. Please elaborate on the discussion.

Thank you very much for your contribution. We also believe that the bias towards women as the main caregivers in this study may reflect and contribute to structural inequalities that affect the financial stability of families. Therefore, in the discussion, we have added the following sentences (page 10, lines 247-251):

“In addition, these caregivers lack free time and often limit their employment to part-time work or even sacrifice their job and opportunities for advancement, affecting family income and financial stability. In this sense, the family's finances fall on the spouse or partner, increasing the economic vulnerability of the household and the economic dependence of women on their partners [30,31,32].”

We have also incorporated the following bibliographical references:

“30. Fujihara, S.; Inoue, A.; Kubota, K.; Yong, K. F. R.; Kondo, K. Caregiver Burden and Work Productivity Among Japanese Working Family Caregivers of People with Dementia. Int J Behav Med 2019, 26, 125–135. https://doi.org/10.1007/S12529-018-9753-9/FIGURES/1

  1. Hailu, G. N.; Abdelkader, M.; Meles, H. A.; Teklu, T. Understanding the Support Needs and Challenges Faced by Family Caregivers in the Care of Their Older Adults at Home. A Qualitative Study. Clin Interv Aging 2024, 19, 481–490. https://doi.org/10.2147/CIA.S451833
  2. Lee, K.; Seo, C. H.; Cassidy, J.; Shin, H. W.; Grill, J. D. Economic hardships of Korean American family caregivers of persons with dementia: a mixed-methods study. Aging Ment Health 2023, 27, 1762–1769. https://doi.org/10.1080/13607863.2022.2122932”.

COMMENT FROM REVIEWER: Authors are also requested to provide data on the socio-economic status of families in this sample population.

Thank you very much for your comment. The study focuses on psychological and social aspects, such as the relationship between depressive symptoms, caregiver strain and social support with dementia grief. The study follows a clear and detailed approach to the psychosocial aspects considered fundamental for understanding the grief process in this context, aiming to delve into the emotional and social dynamics that influence on dementia grief.

Our results show that having a higher level of education may decrease the intensity of dementia grief and it may be because caregivers had more personal resources such as information, social contacts and familiarity with the regulations, and most likely also more financial resources. These factors, such as the economic background, may have an impact on the intensity of dementia grief and it would be interesting to be able to investigate it. However, this study did not collect data on this issue, so it would be impossible to add this analysis now. We thank you for this idea and will take it into account in future developments of this line of research.

COMMENT FROM REVIEWER: Needs mild improvement in the flow of the manuscript.

We are grateful for your comments on the manuscript. We hope that, once the modifications have been made, the article will meet your expectations in improving the fluency of the text to ensure a clearer reading.

Reviewer 3 Report

Comments and Suggestions for Authors

This is a report investigating dementia grief. I think this is a useful paper, as it shows that being female, having a low educational background, and having low social support are associated with depressive symptoms.

However, it is necessary to add comments on the following points.

① Reason for deciding on the number of cases of 250. Have you examined the necessary number through statistical processing in advance?

②In addition to answering the questionnaire, it seems necessary to conduct semi-structured interviews.

③ It is necessary to consider the economic background as a factor.

④It is necessary to perform factor analysis to explain the structure of correlations between observed variables.

Comments on the Quality of English Language

The English text is generally understandable, but there are some grammatical problems.

Author Response

RESPONSE TO REVIEWERS

Dear Reviewer,

Thank you for your interest in our manuscript medicina-2937528 "Relationship between depressive symptoms, caregiver strain, and social support with dementia grief in family caregivers”. We sincerely thank for the editorial and peer review comments that have allowed us to improve the quality of the draft.

We have attached the modified draft and we respond here to the suggestions, including specific references to alterations in the text.

Yours sincerely.

COMMENT FROM REVIEWER: This is a report investigating dementia grief. I think this is a useful paper, as it shows that being female, having a low educational background, and having low social support are associated with depressive symptoms.

Thank you very much. We appreciate your comment and we are pleased that you consider our work adequate.

COMMENT FROM REVIEWER: Reason for deciding on the number of cases of 250. Have you examined the necessary number through statistical processing in advance?

Thank you for your comment. With regard to the sample size, it is true that no statistical calculation has been carried out and we admit that the sample size is insufficient to draw a conclusion.

However, it is important to take into consideration that the study is based on caregivers of people with dementia who face a significant burden of responsibilities. These caregivers are fully dedicated to the care of their loved ones and have very little time to devote to other activities, such as participation in research studies. This sample was all we were able to achieve, as the participants belong to associations in the province of Huelva. Despite this, our sample finally consisted of 250 family caregivers who agreed to participate in the research and who met the inclusion criteria of the study.

Even so, we have added in the limitations section the following sentences (page 11, lines 304-309):

"On the one hand, the sample size is insufficient to draw a conclusion. However, it is important to take into consideration that the study is based on caregivers of people with dementia who face a significant burden of responsibilities. These caregivers are fully dedicated to the care of their loved ones and have very little time to devote to other activities, such as participation in research studies."

COMMENT FROM REVIEWER: In addition to answering the questionnaire, it seems necessary to conduct semi-structured interviews.

Thank you very much for your contribution. We find the idea of exploring the phenomenon of dementia grief from a qualitative perspective very interesting. However, this study was not planned from a qualitative or mixed perspective. Moreover, we do not currently have any qualitative data available to add to the study.

Even so, we are very grateful for the proposal and will consider incorporating qualitative data and analysis in future studies on this line of research, which we will continue to develop.

COMMENT FROM REVIEWER: It is necessary to consider the economic background as a factor.

Thank you very much for your comment. The study focuses on psychological and social aspects, such as the relationship between depressive symptoms, caregiver strain and social support with dementia grief. The study follows a clear and detailed approach to the psychosocial aspects considered fundamental for understanding the grief process in this context, aiming to delve into the emotional and social dynamics that influence on dementia grief.

Our results show that having a higher level of education may decrease the intensity of dementia grief and it may be because caregivers had more personal resources such as information, social contacts and familiarity with the regulations, and most likely also more financial resources. These factors, such as the economic background, may have an impact on the intensity of dementia grief and it would be interesting to be able to investigate it. However, this study did not collect data on this issue, so it would be impossible to add this analysis now. We thank you for this idea and will take it into account in future developments of this line of research.

COMMENT FROM REVIEWER: It is necessary to perform factor analysis to explain the structure of correlations between observed variables.

Thank you very much for your recommendation. Factor analysis is a multivariate statistical method that allows parsimonious explanation of the covariation observed between a set of measured variables, but as a function of one or more constructs or latent variables. That is, the final usefulness of factor analysis is the identification of latent variables (unobserved, also called factors) underlying the observed variables. Thus, a factor can be defined as an unobservable variable that influences more than one observed measure and that explains the correlations between these observed measures (Brown, 2015).

Given the descriptive design of our study, the identification of latent constructs (or unobserved variables) that may underlie the observed variables is not among its objectives.

Furthermore, the number of main variables evaluated (three independent and one dependent) is very limited and they are linearly related to each other (linear dependence).

For all these reasons, we believe that the application of a factor analysis would not provide additional information of interest to the results and conclusions reported in the current version of the manuscript.

Some references:

Brown T. A. (2015). Confirmatory factor analysis for applied research (2nd ed.). New York, NY: Guilford Press.

Watkins, M. W. (2018). Exploratory factor analysis: A guide to best practice. Journal of Black psychology, 44(3), 219-246.

Kline, R. (2013). Exploratory and confirmatory factor analysis. In Applied quantitative analysis in education and the social sciences (pp. 171-207). Routledge.

Bandalos, D. L., & Finney, S. J. (2018). Factor analysis: Exploratory and confirmatory. In The reviewer’s guide to quantitative methods in the social sciences (pp. 98-122). Routledge.

Alhija, F. A. N. (2010). Factor analysis: An overview and some contemporary advances. International Encyclopedia of Education, 3, 162-170. Elsevier.

COMMENT FROM REVIEWER: The English text is generally understandable, but there are some grammatical problems.

We appreciate your comment. Special attention has been paid to grammar and a revision of the text has been made to avoid possible errors. We hope that you will find the text well drafted and comprehensible in its entirety.

Round 2

Reviewer 1 Report

Comments and Suggestions for Authors

The authors address my comments

Reviewer 2 Report

Comments and Suggestions for Authors

Thank you for adding revisions to the manuscript.

Comments on the Quality of English Language

Better